# Estimation of Winter Wheat Yield in Arid and Semiarid Regions Based on Assimilated Multi-Source Sentinel Data and the CERES-Wheat Model

**DOI:** 10.3390/s21041247

**Published:** 2021-02-10

**Authors:** Zhengchun Liu, Zhanjun Xu, Rutian Bi, Chao Wang, Peng He, Yaodong Jing, Wude Yang

**Affiliations:** 1College of Resource and Environment, Shanxi Agricultural University, Taigu 030801, China; lzcsxau@163.com (Z.L.); zjxu163@126.com (Z.X.); hepeng7777@126.com (P.H.); jydyyzsq@163.com (Y.J.); 2National Experimental Teaching Demonstration Center for Agricultural Resources and Environment, Shanxi Agricultural University, Taigu 030801, China; 3College of Agriculture, Shanxi Agricultural University, Taigu 030801, China; wcqxx2005@126.com (C.W.); sxauywd@126.com (W.Y.)

**Keywords:** multi-source Sentinel data, CERES-Wheat model, data assimilation, water stress, yield estimation

## Abstract

The farmland area in arid and semiarid regions accounts for about 40% of the total area of farmland in the world, and it continues to increase. It is critical for global food security to predict the crop yield in arid and semiarid regions. To improve the prediction of crop yields in arid and semiarid regions, we explored data assimilation-crop modeling strategies for estimating the yield of winter wheat under different water stress conditions across different growing areas. We incorporated leaf area index (LAI) and soil moisture derived from multi-source Sentinel data with the CERES-Wheat model using ensemble Kalman filter data assimilation. According to different water stress conditions, different data assimilation strategies were applied to estimate winter wheat yields in arid and semiarid areas. Sentinel data provided LAI and soil moisture data with higher frequency (<14 d) and higher precision, with root mean square errors (RMSE) of 0.9955 m^2^ m^−2^ and 0.0305 cm^3^ cm^−3^, respectively, for data assimilation-crop modeling. The temporal continuity of the CERES-Wheat model and the spatial continuity of the remote sensing images obtained from the Sentinel data were integrated using the assimilation method. The RMSE of LAI and soil water obtained by the assimilation method were lower than those simulated by the CERES-Wheat model, which were reduced by 0.4458 m^2^ m^−2^ and 0.0244 cm^3^ cm^−3^, respectively. Assimilation of LAI independently estimated yield with high precision and efficiency in irrigated areas for winter wheat, with RMSE and absolute relative error (ARE) of 427.57 kg ha^−1^ and 6.07%, respectively. However, in rain-fed areas of winter wheat under water stress, assimilating both LAI and soil moisture achieved the highest accuracy in estimating yield (RMSE = 424.75 kg ha^−1^, ARE = 9.55%) by modifying the growth and development of the canopy simultaneously and by promoting soil water balance. Sentinel data can provide high temporal and spatial resolution data for deriving LAI and soil moisture in the study area, thereby improving the estimation accuracy of the assimilation model at a regional scale. In the arid and semiarid region of the southeastern Loess Plateau, assimilation of LAI independently can obtain high-precision yield estimation of winter wheat in irrigated area, while it requires assimilating both LAI and soil moisture to achieve high-precision yield estimation in the rain-fed area.

## 1. Introduction

Monitoring crop growth and estimating yield are crucial for agricultural management and national food security [1,2,3]. Assimilation of remote sensing data into crop models is an effective approach for estimating or predicting crop yield at a regional scale [4,5,6,7,8]. The accuracy of parameters derived from remote sensing data has an important impact on the estimation accuracy, which depends on the source of data from remote sensing and the inversion algorithm [9,10,11,12]. Pauwels et al. [13] found that assimilating remote sensing data into crop models significantly improved the accuracy of the yield estimation if the error of the leaf area index (LAI) derived from remote sensing data was <1 m^2^ m^−2^ or the error in soil moisture was <0.05 cm^3^ cm^−3^, and the temporal resolution was <14 d. In a study based in northern France, Casa et al. [14] assimilated measured LAI, rather than LAI derived from remote sensing data, into the STICS model. They reported better dynamic adjustment of LAI with longer observation periods and increased the amount of observational data by adjusting the input frequency of the measured LAI data. Curnel et al. [15] assimilated LAI derived from SPOT-HRV into the WOFOST model, adjusted the input frequency of the LAI, and found that the accuracy of the yield estimation was highest when the time-frequency of the LAI was 3–7 d. Previous studies showed that the assimilation of parameters derived from high spatial-temporal resolution remote sensing data into crop models improved the accuracy of yield estimation at regional scales.

Currently, remote sensing data that are commonly used to derive LAI and soil moisture are MODIS-LAI products, soil moisture products (AMSR-E soil moisture product, Ocean Salinity (SMOS) mission), and LAI and soil moisture derived from Landsat images. The MODIS-LAI products have a spatial resolution of 250–1000 m. The low spatial resolution in MODIS products results in a large number of mixed pixels. The MODIS-LAI is lower than the LAI measured in the field [16]. Soil moisture product data (AMSR-E soil moisture product, Ocean Salinity (SMOS) mission) has a spatial resolution of 25 km and a low product accuracy, which is the result of the crop cover on the surface during vigorous crop growth [13,17,18]. Both LAI and soil moisture derived from Landsat images have a spatial resolution of 30 m and a temporal resolution of 16 d. Landsat products provide limited information during the winter wheat growth period due to low temporal resolution and the susceptibility of optical remote sensing to interference by rain and snow. The inputs from low-frequency remote sensing images do not greatly modify the simulated LAI or simulated soil moisture profiles of the crop model, limiting the improvements in estimation accuracy of the crop model through assimilation [15]. However, the data of the Earth Observation Satellite Sentinel from the European Space Agency are ideal for assimilation with crop models because of the high spatial and temporal resolution [19,20]. At present, researchers have tried to assimilate Sentinel-2 data to crop models for yield estimation since this method has good application potentials [21,22].

Data assimilation algorithms play a central role in integrating remote sensing data into crop models. Parameter optimization methods that are based on cost functions and filtering methods from estimation theory are two main types of data assimilation methods. In parameter optimization methods, we iteratively adjust the parameters or initial conditions to minimize the difference between remote sensing observations and model simulation values. The parameters or initial conditions are closely related to growth and yield formation in crop models, and they are difficult to obtain by conventional methods [4,9]. Curnel et al. [15] used the four-dimensional variational algorithm to assimilate the LAI derived from remote sensing data and WOFOST models to optimize the input parameters of the WOFOST model. The core of the filtering algorithm is to integrate remote sensing observation data from different resolutions within the dynamic framework of the mechanism process model so that the mechanism process model and various observation operators can become a prediction system that continuously relies on external observations to automatically adjust the model trajectory and reduce errors. de Wit et al. [23] used the ensemble Kalman filter (EnKF) algorithm to assimilate the soil moisture derived from microwave remote sensing and the WOFOST model to correct the water balance process of the crop model.

For parameter selection in assimilation, it is a common practice to assimilate a single variable into crop models to improve the accuracy of crop yield estimation at a regional scale. Some studies also assimilated two variables into crop models. For example, de Wit et al. [23] and Nearing et al. [24] assimilated two variables (i.e., LAI and soil moisture, or LAI and evapotranspiration (ET)) into crop models. They found that assimilation with two variables improved the accuracy of estimating crop yields at a regional scale compared with the method that only assimilated a single variable because it corrected for canopy growth and soil water balance. However, further studies by Ines et al. [18] and Mishra et al. [25] found that assimilation of two variables (i.e., LAI and soil moisture) was suitable for predicting yield in years with average precipitation, although assimilation of a single variable (i.e., LAI) into crop models was more suitable for predicting yield in extremely wet years. In arid and semiarid areas, precipitation is limited. Not all croplands have irrigation conditions. Some areas can be irrigated, but other arid crop areas can not, and the growth of crops in those areas are affected by water stress. By selecting optimal parameters from remote sensing information and assimilating those parameters into crop models, we could improve the prediction of crop yields across different regions, which would aid in decisions about the allocation of water in agricultural areas.

The aims of this study were (1) Using LAI and soil moisture derived from Sentinel images to provide high temporal and spatial resolution data for data assimilation-crop modeling. (2) Using ensemble Kalman filter data assimilation to modify the LAI and soil moisture simulated by the CERES-Wheat model. (3) Seeking the optimal data assimilation-crop modeling strategies for estimating the yield of winter wheat under different water stress conditions.

## 2. Materials and Methods

### 2.1. Study Area

The research areas are located in the southeast of the Loess Plateau in China and included three counties: Xiangfen county, Xinjiang county, and Wenxi county (110°59′33″—111°40′31″ E, 35°9′38″—36°03′14″ N, altitude between 184–1535 m) (Figure 1a). The study area is characterized by a warm-temperate, continental, monsoon climate with an average annual rainfall of 450–600 mm, an annual average temperature of 12–14 °C, and a frost-free period of 160–190 d. The crops in this region are mainly wheat in winter and maize in summer. As the main crop in the study area, the planting area of winter wheat is 108,061 ha, which accounts for 73.31% of the farmland area.

### 2.2. Data

#### 2.2.1. Field Measurements

There were 45 sampling sites across the study area (Figure 1b). Measurements from 20 sites (red points in Figure 1b) were used to parameterize the assimilation for crop yield estimation, and measurements from 25 sites (yellow points in Figure 1b) were used to verify the accuracy of yield estimation of winter wheat. In September 2017, 20 sites (red points) were selected for soil sampling. Soil samples were taken for physical and chemical analyses before winter wheat sowing. The soil profile was divided into 7 layers (0–10 cm, 10–20 cm, 20–50 cm, 50–80 cm, 80–120 cm, 120–160 cm, and 160–200 cm). The physical and chemical properties of each layer of the soil profile were measured in the laboratory. LAI and soil moisture at all sites were measured during the jointing stage on 19 March 2018 and 17 March 2019, and during the heading-filling stage on 16 April 2018 and 19 April 2019. LAI was measured using dry weight. The measurement was sampled within a 1 m × 1 m area. Thirty winter wheat samples were selected from the total sample, and then the stems and leaves were separated. We took the leaves and measured the length and width of each leaf (the widest part of the leaf). We multiplied the length and width of each leaf and summed all 30 samples of winter wheat to determine the total leaf area. The leaf samples were dried and weighed again to obtain the leaf dry weight. Combined with the total leaf area, we calculated the leaf area corresponding to the dry weight per unit leaf area. Based on the total number of the samples, the dry leaf weight of 30 wheat plants, and the leaf area corresponding to the dry weight per unit leaf area, we calculated the total canopy area of the sampling area. The LAI was calculated based on the sampling area and the total canopy area of the sampling area. The moisture content of the 0–20 cm soil depth was measured using the drying method. During the mature stage of winter wheat in 2018 and 2019, the yield was measured within a 1 m × 1 m area. Samples were dried, threshed, and weighed to calculate the yield (kg ha^−1^). In addition, field management data were collected from 2017–2019. This included wheat varieties, planting dates, maturity dates, planting methods, planting density, fertilization dates and amounts, and irrigation dates and amounts. In this study, the measured and field management data from 2018 were used to calibrate the CERES-Wheat model. Data in 2019 were used to conduct assimilation to estimate the regional yield of winter wheat in the study area.

#### 2.2.2. Multi-Source Sentinel Data

Sentinel-1 and Sentinel-2 belong to a series of satellites from the European Space Agency (ESA). Sentinel-1 is a radar satellite, and Sentinel-2 is an optical satellite. Sentinel-1 is equipped with a C-band synthetic aperture radar (SAR), which is not constrained by light and weather conditions, and provides continuous images at all times under any weather conditions. It has a spatial resolution of 5 m × 20 m and a temporal resolution of 12 d [26]. The temporal resolution of Sentinel-2 optical images is 5 d. The spatial resolution of the red and near-infrared bands that were used to calculate the vegetation index is 10 m [27]. The combination of the two satellites monitored a large area and provided data with a high revisit period and high resolution, which sufficiently met the data requirements for data assimilation between remote sensing and crop models [4]. During the main growth stages of winter wheat (from green-up stage to milking stage), Sentinel-1 images were selected at seven dates: 2 March, 14 March, 26 March, 7 April, 19 April, 1 May, and 13 May in 2019. The Sentinel-2 images with <10% cloud content were selected at seven dates: 17 March, 1 April, 16 April, 11 May, 21 May, 26 May, and 10 June in 2019.

### 2.3. Extraction of Irrigated and Rain-Fed Winter Wheat Planting Areas

#### 2.3.1. Extraction of Winter Wheat Planting Areas

Types of land use in the study area included farmland, forest, grassland, and construction land. Farmland was mainly planted with winter wheat and maize. Combined with the different phenological characteristics between winter wheat and other vegetation with spectral differences at different times, we extracted the winter wheat planting area by using a decision tree method. Based on the decision tree method proposed by Li et al. [28], normalized difference vegetation index (NDVI) was calculated using three Sentinel-2 images on 17 March, 16 April, and 10 June 2019. Based on the manual discrimination and the comparison with the distribution of farmland in the farmland fertility database of Shanxi Province in 2014, we determined the extracted NDVI threshold from winter wheat as 0.25. This means that the NDVI threshold on 17 March and 16 April was greater than 0.25, while the NDVI threshold on 10 June was less than 0.25. The NDVI value on 16 April was greater than on 10 June. The extracted winter wheat area was 103,254 ha, which was consistent with the statistics of the Shanxi Provincial Agriculture Bureau (108,061 ha). The extraction accuracy of the winter wheat planting area was 96.0%.

#### 2.3.2. Irrigated Areas and Rain-Fed Areas for Winter Wheat

According to different water stress conditions, the area covered with winter wheat was divided into irrigated areas (non-water stressed) and rain-fed areas (water-stressed). Irrigated areas were <600 m in altitude, and the slope was <15°. Rain-fed areas had an elevation >600 m and a slope >15° [29]. Using the above thresholds, we used a decision tree to define irrigated areas and rain-fed areas in the research area, and we calculated the corresponding planting area (Figure 2). The area of irrigated land in winter wheat was 82,913 ha, and the area of rain-fed land in winter wheat was 20,341 ha, which accounted for 80.3% and 19.7%, respectively, of the total area planted with winter wheat. We compared the irrigated areas for winter wheat extracted from the decision tree with the extracted areas from the farmland fertility database of Shanxi Province in 2014 (Figure 3). The extraction accuracy was 81%, suggesting a reliable estimation of irrigated and rain-fed areas for winter wheat.

### 2.4. Water Cloud Model

In this study, soil moisture in the winter wheat planting areas was derived from Sentinel-1 radar data using the water cloud model. In 1978, Attema and Ulaby (1978) proposed a water cloud model to estimate soil moisture in agricultural lands. In this model, they assumed that the crop canopy was a homogeneous scatterer. They described the microwave backscatter of the crop canopy as the sum of two parts (i.e., the volume scattering obtained by direct reflection of crops and the surface scattering after dual attenuation of crops) [30]. The model is expressed as Equation (2):(1)$$\sigma_{can}^{0}(\theta )=\sigma_{veg}^{0}(\theta )+\gamma^{2}(\theta )\cdot \sigma_{soil}^{0}(\theta )$$
(2)$$\sigma_{veg}^{0}(\theta )=A\cdot m_{veg}\cdot \cos(\theta )\cdot (1-\gamma^{2}(\theta ))$$
(3)$$\gamma^{2}(\theta )=\exp(-2\cdot B\cdot m_{veg}\cdot \sec(\theta ))$$
where $\sigma_{can}^{0}(\theta )$ is the total microwave backscattering coefficient of the croplands, $\sigma_{veg}^{0}(\theta )$ is the backscattering coefficient of the crop canopy, $\sigma_{soil}^{0}(\theta )$ is the soil direct backscattering coefficient, *r*^2^*(θ)* is the dual attenuation factor of microwave penetration through the canopy, *θ* is the incidence angle of a microwave, *m_veg_* is the water content of plants (kg m^−2^), and A and B are parameters dependent on vegetation types. Bindlish and Barros (2001) calculated parameters of the water cloud model in lands with different covers and found that the canopy parameters of winter wheat were *A* = 0.0018 and *B* = 0.138 [31].

Plant water content, *m_veg_*, is an important input parameter of the water cloud model. There is a good quantitative relationship between *m_veg_* and the spectral index. This is the theoretical basis for calculating plant water content using the empirical model [32]. According to Jackson et al. [33], the relationship between *m_veg_* and normalized difference water index (NDWI) is as follows:(4)$$m_{veg}=1.44\cdot NDWI^{2}+1.36\cdot NDWI+0.34$$
(5)NDWI=(ρ(NIR)-ρ(MIR))/(ρ(NIR)+ρ(MIR))

### 2.5. CERES-Wheat Model

CERES-Wheat is one of the sub-models in the DSSAT (Decision Support System for Agrotechnology Transfer) model series, which was developed especially for wheat. The model simulates the growth and development of wheat, yield, and nitrogen-carbon-water balance at a daily scale by using meteorological and soil databases and modules of soil moisture, nitrogen, and carbon balance [34,35]. The required data for running the CERES-Wheat model included four parts: meteorological data, soil information, field management, and genetic parameters of crops [34]. The meteorological data included the daily maximum and minimum temperatures, the daily total precipitation, and the daily integrated solar radiation at the research sites. These data were downloaded from the China Meteorological Administration (CMA) database (http://data.cma.cn/ (accessed on 1 June 2019). Soil information included clay content, silt content, bulk density, field water holding capacity, saturated soil water content, organic carbon content, total nitrogen, available potassium, available phosphorus, cation exchange capacity, and pH, which were obtained through field measurements. Field management included planting dates, planting density, fertilization dates and amounts, and irrigation dates and amounts. In the CERES-Wheat model, there are seven genetic parameters of winter wheat. Genetic parameters control the growth and development of wheat and are directly related to the development of plant morphology and crop yield. Therefore, genetic parameters need to be calibrated before any application of the model [23,34,36]. The calibration of genetic parameters generally uses the ‘trial-and-error’ method [14,28,37,38]. The meteorological, soil, and field management data for winter wheat were used as the input parameters for the CERES-Wheat model during the growth stages during 2017–2018. The measured LAI, yield, and harvest dates during the growth stages were used to calibrate the model to determine the genetic parameters of winter wheat in the study area. RMSE between the simulated LAI of the calibrated CERES-Wheat model and the measured LAI was 1.1243 m^2^ m^−2^. RMSE of yield between predictions and measurements was 622.23 kg ha^−1^. The difference in harvest date between predictions and measurements was <4 d. Therefore, the accuracy of the calibrated CERES-Wheat model was high.

### 2.6. Ensemble Kalman Filter (EnKF) Assimilation Algorithm

For the implementation of the EnKF, we based our research on the work of Evensen [39] and Xie et al. [38]. The basic analysis steps in an EnKF for each ensemble member was defined as
$$A_{k}^{f}=M(A_{K-1}^{a})$$
(6)$$A_{k}^{a}=A_{k}^{f}+P_{k}H^{T}{\left(HP_{k}H^{T}+R_{k}\right)}^{-1}(D_{k}-HA_{k}^{f})$$
where $A_{k}^{a}$ and $A_{k}^{f}$ are the analyzed and forecasted matrices, respectively, of ensemble states at time *k*, *P_k_* and *R_k_* are the ensemble and observation covariance matrices, respectively, *H* is the measurement operator, and $D_{k}-HA_{k}^{f}$ are the innovation vectors. In this study, LAI and soil moisture were treated as state variables, and the observations consisted of Sentinel-retrieved LAI and Sentinel-retrieved soil moisture.

Assimilation followed two steps. First, the initial parameters were input to the CERES-Wheat model to simulate the growth of winter wheat. Second, based on the measured LAI, the standard errors of simulated LAI from the CERES-Wheat model and the derived LAI from the Sentinel data were calculated. The same process was applied to the soil moisture data. Based on the standard deviations between simulated data and measurements, the standard deviations were set to 17% and 9% for the simulated LAI and simulated soil moisture of the CERES-Wheat model, respectively. Similarly, the standard deviations of LAI and soil moisture derived from multi-source Sentinel data were set to 13% and 8%, respectively. Then, at the initial time k of the EnKF assimilation, the Monte Carlo method was used to perturb the LAI and soil moisture simulated by the CERES-Wheat model to generate a forecast set of state variables based on the standard deviations of the simulated state variables. The measured LAI and soil moisture were perturbed to generate a data set of the same size as the observed set according to standard deviations of state variables derived from remote sensing. The forecast set and the observed set were substituted into the assimilation to produce the assimilated LAI and the assimilated soil moisture at time *k*, which were used to predict the LAI and soil moisture at time *k* + 1.

## 3. Results

### 3.1. LAI Derived from Sentinel-2

There is a good quantitative relationship between the NDVI and LAI [40,41,42]. First, we calculated the NDVI from the seven selected Sentinel-2 images. Then, we obtained the corresponding NDVI of the Sentinel-2 image on 17 March 2019 for the 45 locations in the field based on the geographic coordinates. We established a regression model between NDVI and LAI from field measurements on the same day (Equation (1)). The regression model had a determination coefficient (R^2^) of 0.52 and a root mean square error (RMSE) of 0.9955 m^2^ m^−2^ and was statistically significant (*p* < 0.001). LAI from the seven Sentinel-2 images was derived according to Equation (7):(7)LAI=8.8049*NDVI−0.9866

### 3.2. Soil Moisture Derived from Sentinel-1

After separating the contribution of vegetation scattering and absorption from the total backscatter of the microwave by using the water cloud model, the direct backscatter of the soil was obtained as $\sigma_{soil}^{0}(\theta )$. Then, we matched the corresponding $\sigma_{soil}^{0}(\theta )$ from Sentinel-1 imaging pixels on 19 April 2019 for the locations measured in the field based on the geographic coordinates. We further established a regression model between $\sigma_{soil}^{0}(\theta )$ and the measured soil moisture in the field on 19 April 2019 (Equation (8)). The regression model had a determination coefficient (R^2^) of 0.47, an RMSE of 0.0305 cm^3^ cm^−3^, and was statistically significant (*p* < 0.01). The soil moisture of the seven selected Sentinel-1 images was calculated according to Equation (8):(8)$$Soilwater=0.0133\sigma_{soil}^{0}(\theta )+0.3815$$

### 3.3. Analysis of Assimilated LAI

The LAI derived from Sentinel-2 data and the LAI simulated by the CERES-Wheat model were assimilated using the EnKF algorithm to obtain the assimilated LAI. The assimilated LAI maintained the same patterns as the simulated LAI. The LAI increased rapidly from the green-up stage to the heading-filling stage and reached the maximum at about 200 d after sowing. The assimilated LAI and the simulated LAI reached their maximum values at the same time. Then, LAI began to decrease. During the green-up stage to the heading-filling stage of winter wheat, the simulated LAI was lower than the measured LAI. Using LAI derived from remote sensing to correct the simulation profile, the value of the assimilated LAI increased substantially and was closer to the measurements in the field. After the LAI peak, the simulated LAI decreased rapidly. Using LAI derived from remote sensing to correct the simulated LAI, the decline in the assimilated LAI slowed down, which was more in line with the actual change of LAI during the winter wheat filling stage (Figure 4).

To test the accuracy of simulated LAI and assimilated LAI, we used field-measured LAI from 20 assimilation sampling sites and conducted linear regressions between the measured LAI and the simulated LAI and the assimilated LAI. We calculated the RMSE and the corresponding determination coefficient (R^2^). All regressions were significant (*p* < 0.001). The regression between assimilated LAI and measured LAI had a greater R^2^ (0.6446) than the regression between assimilated LAI and simulated LAI (R^2^ = 0.6186). The assimilated LAI had a smaller RMSE (1.1886 m^2^ m^−2^) than the simulated LAI’s RMSE (1.6344 m^2^ m^−2^) (Figure 5). The accuracy of the assimilated LAI was higher than that of the simulated LAI, and it was more representative of the actual situation of winter wheat LAI.

To estimate the regional yield of winter wheat using assimilated LAI, we extended the LAI assimilation values from the field scale to the regional scale. Considering factors, such as the phenological period of winter wheat and the imaging time of the remote sensing images, we selected the derived LAI from the Sentinel-2 images on 17 March, 1 April, 16 April, and 21 May in 2019, as the LAI at the green-up, jointing, heading-filling, and milking stages, respectively. Regression analysis was performed on the selected LAI and the assimilated LAI at the corresponding dates at the 20 sample sites to obtain the assimilated LAI at the regional scale across the phenophase.

### 3.4. Analysis of the Assimilated Soil Moisture

The soil moisture values derived from the Sentinel-1 data and the soil moisture values simulated by the CERES-Wheat model were assimilated using the EnKF algorithm to obtain the soil moisture assimilation values for growth stages of winter wheat at a daily scale. Taking three irrigated areas, which included Nanxindian village in Xiangfen county, Zezhang village in Xinjiang county, and Su village in Wenxi county, and three rain-fed areas, which included Dongguo village in Xiangfen county, Bolin village in Wenxi county, and Hutou village in Wenxi county, as examples, we compared the assimilated soil moisture, the simulated soil moisture, and soil moisture derived from Sentinel-1 (Figure 6). The assimilated soil moisture maintained the trend change characteristics of soil moisture simulated by the CERES-Wheat model. Using soil moisture derived from Sentinel-1 to correct the simulation profile, the assimilated soil moisture profile of the sampling sites in irrigated areas was lower than simulations. The assimilated soil moisture values of the sampling sites in rain-fed areas increased from day 150 to 190. From day 190 to 250, the assimilated soil moisture profile decreased compared with the simulated profile, which was closer to the measurements.

To test the accuracy of simulated soil moisture and assimilated soil moisture, we used field-measured soil moisture from 20 assimilation sampling sites to conduct linear regressions between the simulated soil moisture and the assimilated soil moisture against the measured soil moisture. We calculated the RMSE and R^2^. All regressions were significant (*p* < 0.01). The assimilated soil moisture had a greater R^2^ (0.6734) than the simulated soil moisture’s R^2^ (0.6547). The assimilated soil moisture had a smaller RMSE (0.0412 cm^3^ cm^−3^) than the simulated soil moisture’s RMSE (0.0656 cm^3^ cm^−3^) (Figure 7). The accuracy of the assimilated soil moisture was higher than that of the simulated soil moisture.

To estimate the regional winter wheat yield using assimilated soil moisture, we extended assimilated field-scale soil moisture values to the regional scale. Considering factors, such as the phenological period of winter wheat and the imaging time of remote sensing images, we selected the derived soil moisture from the Sentinel-1 images on 2 March, 7 April, 19 April, and 13 May in 2019, as the soil moistures for the green-up, jointing, heading-filling, and milking stages, respectively. Regression analysis was performed on the selected soil moisture values and the assimilated soil moisture values for the corresponding dates from 20 sampling sites to obtain the assimilated soil moisture at the regional scale across the phenophase.

### 3.5. Selection and Analysis of Assimilation Variables in Yield Estimation

To select suitable assimilation strategies for rain-fed areas of winter wheat and irrigated areas that would estimate yield with high accuracy, we used three different assimilation strategies: (1) assimilate LAI alone to estimate winter wheat yield, (2) assimilate soil moisture alone to estimate winter wheat yield, and (3) assimilate LAI and soil moisture simultaneously to estimate winter wheat yield. We used these three strategies for the estimation of winter wheat yields in different regions to determine the assimilation estimation strategy that was applicable to irrigated areas and rain-fed areas.

In this study, we used the assimilated LAI, assimilated soil moisture, or assimilated LAI and soil moisture to construct assimilation models to estimate winter wheat yields. According to the analytical hierarchy, the weights of LAI that contributed to the yield of winter wheat at the green-up, jointing, heading-filling, and milking stages were 0.0550, 0.2650, 0.5660, and 0.1140, respectively, and the weights for soil moisture were 0.0555, 0.5655, 0.2605, and 0.1185, respectively. Based on these weights, the weighted assimilated LAI, the weighted assimilated soil moisture, or the weighted assimilated LAI and soil moisture in the main growth stages of winter wheat were calculated to conduct linear regressions with the measured winter wheat yield.

In irrigated areas for winter wheat, 10 sample sites were used to establish the yield estimation model. The remaining 15 sites were used to verify the accuracy of the yield. Ten sites in the rain-fed areas were used to establish the corresponding estimation model, and the remaining 10 sites were used for verification (Table 1 and Table 2). The yield estimation model constructed either by assimilated LAI independently or by assimilated both LAI and soil moisture in irrigated areas for winter wheat showed good performance. The R^2^ of the two methods was the same, although the RMSE and absolute relative error (ARE) from the method with assimilated LAI were smaller than that of the assimilated LAI and soil moisture. The yield estimation model constructed by assimilated soil moisture independently had the lowest R^2^ and the largest RMSE and ARE. Therefore, the application of assimilated LAI independently estimated the yield of winter wheat with high accuracy in irrigated areas and improved the efficiency in yield estimation. In rain-fed areas, the yield estimation model constructed by the assimilated both LAI and soil moisture had the highest R^2^ and the smallest RMSE and ARE. The model constructed by the assimilated soil moisture had a lower R^2^ and higher RMSE and ARE, but the model constructed by the assimilated LAI had the lowest R^2^ and the largest RMSE and ARE.

For the yield estimation of winter wheat across the study area, the model constructed by assimilating LAI alone was used in irrigated areas, and the model constructed by assimilated LAI and soil moisture simultaneously was used in rain-fed areas (Figure 8). Due to water stress, the yield in rain-fed areas was significantly lower than in the irrigated areas.

## 4. Discussion

### 4.1. Effects of Remote Sensing Data with High Spatial-Temporal Resolution from Multiple Sources on the Accuracy of Assimilation Parameters

A previous study showed that the error from remotely sensed LAI was <1 m^2^ m^−2^, and the error from remotely sensed moisture was <0.05 cm^3^ cm^−3^. When the frequency was <2 wk, the accuracy of yield estimation using data assimilation-crop modeling improved significantly [13]. Therefore, the assimilation of remote sensing data with high spatial-temporal resolution into crop models improves the estimation accuracy of crop yield significantly [15].

In this study, we used multi-source Sentinel data to derive LAI and soil moisture. The Sentinel-1 radar data had a temporal resolution of 12 d and a spatial resolution of 5 m × 20 m. It was not affected by cloudy and rainy weather because of its ability to penetrate through those conditions. Highly accurate soil moisture values were derived without the effects of vegetation coverage based on Sentinel 1 data and the water cloud model [41,43]. The red and infrared bands of Sentinel-2 images that were used to derive LAI had a spatial resolution of 10 m and a temporal resolution of 5 d. Compared with the commonly used LAI derived from MODIS or Landsat products, LAI derived from Sentinel-2 images had higher accuracy [40,41,44].

In addition, in this study, we chose seven Sentinel-1 images during the main winter wheat growth stages (from the green-up stage to the maturity stage). The accuracy of the derived soil moisture was RMSE = 0.0305 cm^3^ cm^−3^. We only obtained seven images in total from Sentinel-2 images with cloud content <10%. The RMSE of the derived LAI was <1 m^2^ m^−2^. This met the requirements proposed by Pauwels [13] for the accuracy and frequency of remote sensing information in “estimating crop yield based on remote sensing information and crop model assimilation”.

### 4.2. The Effects of Applying Different Assimilation Strategies on the Prediction Accuracy of Crop Yield in Arid and Semi-Arid Regions

Many researchers have used MODIS and Landsat image to estimate crop yields and achieved high yield estimation accuracy [45,46,47,48,49]. Becker et al. [45] used MODIS data and applied a generalized regression-based model to predict winter wheat yields in Kansas and Ukraine, with an error of 7 to 10%, compared with statistical yields. Ahmad et al. [49] used Landsat imagery and applied machine learning algorithms to predict and assess the interannual variability of maize in Pakistan. The accuracy of the estimate was greater than 90%. In this study, LAI and soil moisture obtained from assimilation Sentinel images with the CERES-Wheat model were used to construct the yield estimation model.

Comparative analysis of the accuracy of the yield estimation models constructed using different assimilation strategies suggested that using assimilated LAI independently in irrigated areas produced the highest accuracy of winter wheat yield estimations (RMSE = 427.57 kg ha^−1^, ARE = 6.07%) (Table 1). For rain-fed areas, using assimilated LAI and soil moisture simultaneously had the most accurate yield estimates (RMSE = 424.75 kg ha^−1^, ARE = 9.55%). In the irrigated areas, LAI assimilation led to enhanced canopy growth. At the same time, because of the abundant availability of soil moisture, the assimilated yield increased, and the estimated yield accuracy was high (RMSE = 427.57 kg ha^−1^, ARE = 6.07%). When assimilating soil moisture, winter wheat growth was not water-limited. Therefore, the improvement in winter wheat canopy growth was limited, which resulted in little improvement in yield simulation and poor accuracy of yield estimation (RMSE = 533.64 kg ha^−1^, ARE = 8.49%). The assimilation of LAI and soil moisture improved the winter wheat canopy growth and adjusted the soil water balance. The yield estimation accuracy increased (RMSE = 436.71 kg ha^−1^, ARE = 6.16%) but was slightly lower than that of the assimilated LAI method.

For rain-fed winter wheat areas, assimilating LAI alone did not capture the increase in water demand caused by the increase in assimilated LAI due to low soil water content in the root zone. Not only did LAI assimilation not increase simulated yield, but it also increased water stress, which resulted in poor accuracy in yield estimates (RMSE = 612.93 kg ha^−1^, ARE = 12.47%). Assimilated soil moisture reduced water stress and increased soil moisture near the surface. It also had a better correction on the soil water balance process, which resulted in high accuracy of yield estimation (RMSE = 467.37 kg ha^−1^, ARE = 11.44%). When assimilating both LAI and soil moisture, assimilated soil moisture reduced the simulated crop water stress, and assimilated LAI corrected the LAI profile, which resulted in the most accurate yield estimate (RMSE = 424.75 kg ha^−1^, ARE = 9.55%).

In the arid and semi-arid regions of the southeastern Loess Plateau, assimilation of LAI independently produced high-precision yield estimation of winter wheat in irrigated areas, although it required assimilation of LAI and soil moisture simultaneously to achieve high-precision yield estimation in rain-fed areas. Ines et al. [18] found that assimilating LAI alone into a crop model produced highly accurate yield estimates for maize in years with abundant precipitation (under extremely humid conditions), but in the years with normal precipitation, the model needed assimilation of both LAI and soil moisture to be accurate. Mishra et al. [25] confirmed that estimating agricultural production in arid regions needed to assimilate both soil moisture and LAI into the crop model framework. That is, water stress limits the choice of assimilation variables. Therefore, the selection of assimilation variables in the assimilation estimation of remote sensing information and crop models should consider whether or not the soil in the area is water-stressed.

## 5. Conclusions

In this study, we chose three counties that grew winter wheat in the southeast of the arid and semiarid Loess Plateau in China as the study area. We assimilated the multi-source Sentinel remote sensing data that has a high spatial and temporal resolution (i.e., Sentinel-1 radar data were used for deriving soil moisture, Sentinel-2 data were used for deriving LAI) into the CERES-Wheat model. This model was used to estimate winter wheat yields by developing different assimilation strategies in different areas that had different water stress conditions.

We have three main conclusions:(1)The RMSE of LAI derived from Sentinel-2 was 0.9955 m^2^ m^−2^, and the RMSE of soil moisture derived from Sentinel-1 was RMSE = 0.0305 cm^3^ cm^−3^. Sentinel data provided high temporal and spatial resolution for deriving LAI and soil moisture in the study area.(2)The advantages of the CERES-Wheat model in temporal continuity and remote sensing in spatial continuity were integrated by the assimilation method using Sentinel data and the CERES-Wheat model. The RMSE of LAI and soil water obtained by the assimilation method was lower than those simulated by the CERES-Wheat model, which were reduced by 0.4458 m^2^ m^−2^ and 0.0244 cm^3^ cm^−3^, respectively.(3)LAI in the irrigated areas of winter wheat fully described the growth and development of the canopy. The assimilation of LAI alone produced high-precision yield estimation in irrigated areas (RMSE = 427.57 kg ha^−1^, ARE = 6.07%). Because of the water stress on the growth of winter wheat in rain-fed areas, assimilation of LAI and soil moisture simultaneously adjusted the growth and development of the canopy and promoted soil water balance and, therefore, produced accurate estimates of yield (RMSE = 424.75 kg ha^−1^, ARE = 9.55%).

In further studies, we will compare the accuracies of yield estimation, assimilating the Sentinel, Landsat, and MODIS into the crop model. We will also explore the use of the Google Earth Engine (GEE) platform in conducting research on remote sensing and crop model data assimilation systems, aiming to improve the calculation efficiency of remote sensing assimilation models at a regional scale.

## Figures and Tables

**Figure 1 sensors-21-01247-f001:**
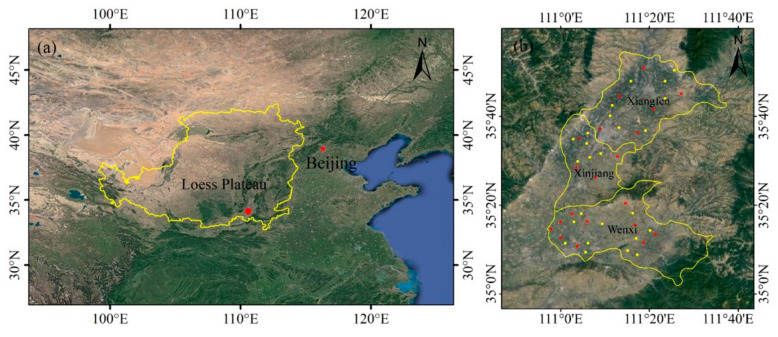
The geographical location and sampling sites of the study area. (**a**) The location of the study area in the Loess Plateau; (**b**) The location of the field measured points in the three counties. All points (including red and yellow points) are field measured points, and the red points are sampling sites for data assimilation.

**Figure 2 sensors-21-01247-f002:**
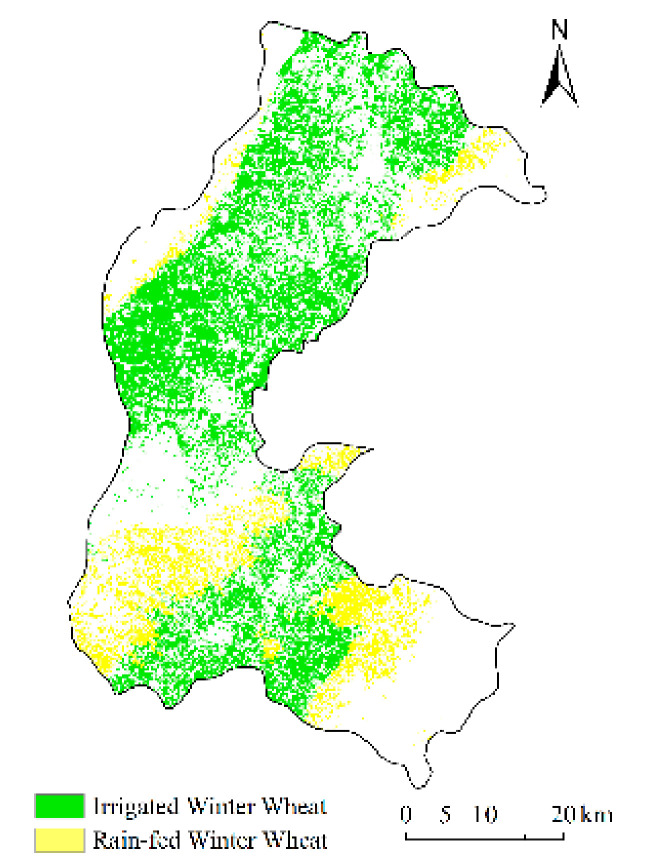
Spatial distribution of irrigated areas and rain-fed areas for winter wheat in Xiangfen, Xinjiang, and Wenxi county.

**Figure 3 sensors-21-01247-f003:**
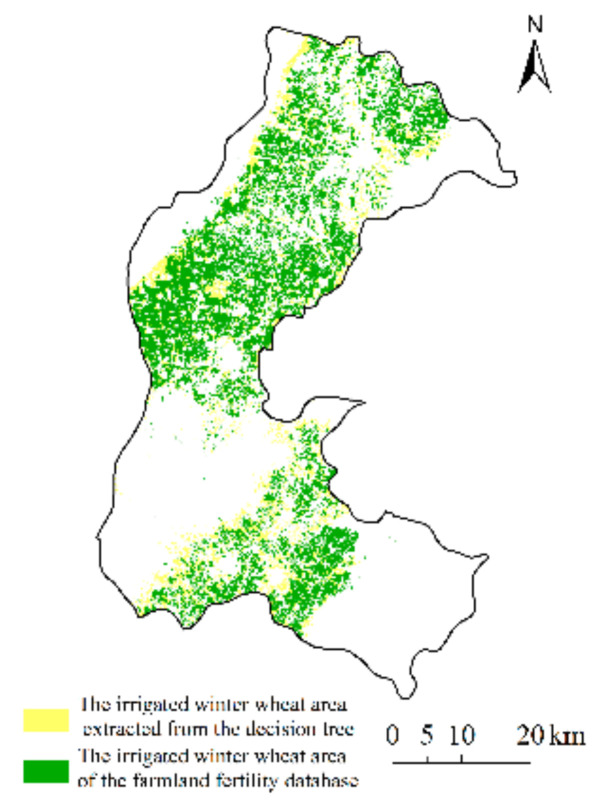
The irrigated winter wheat area extracted from the decision tree and the irrigated winter wheat area of the farmland fertility database of Shanxi Province in 2014.

**Figure 4 sensors-21-01247-f004:**
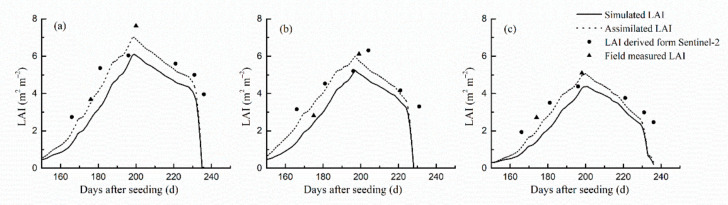
Simulated leaf area index (LAI), assimilated LAI, LAI derived from Sentinel-2, and field-measured LAI for winter wheat in the three sites in 2019. (**a**) Nanxindian village of Xiangfen county; (**b**) Zezhang village of Xinjiang county; (**c**) Su village of Wenxi county in 2019.

**Figure 5 sensors-21-01247-f005:**
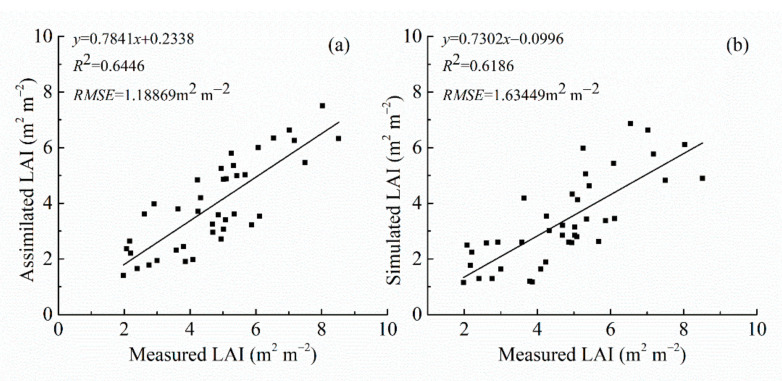
Linear regression analysis with measured LAI. (**a**) Assimilated LAI; (**b**) Simulated LAI.

**Figure 6 sensors-21-01247-f006:**
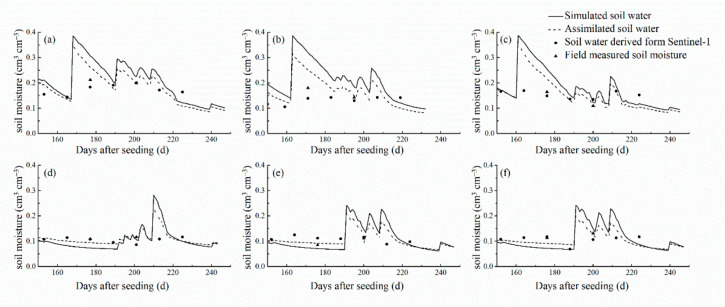
The pattern of simulated soil moisture, assimilated soil moisture, and soil moisture derived from Sentinel-1 from the irrigated areas of winter wheat in (**a**) Nanxindian village of Xiangfen county, (**b**) Zezhang village of Xinjiang county, and (**c**) Su village of Wenxi county and in the rain-fed areas in (**d**) Dongguo village of Xiangfen county, (**e**) Bolin village of Wenxin county, and (**f**) Hutou village of Wenxi county in 2019.

**Figure 7 sensors-21-01247-f007:**
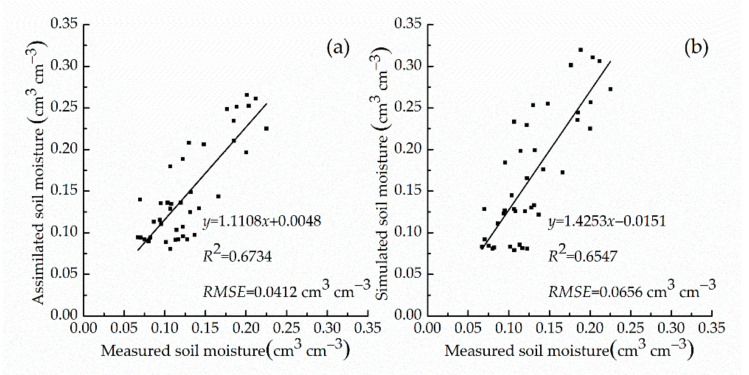
Linear regression analysis with measured soil moisture. (**a**) Assimilated soil moisture; (**b**) Simulated soil moisture.

**Figure 8 sensors-21-01247-f008:**
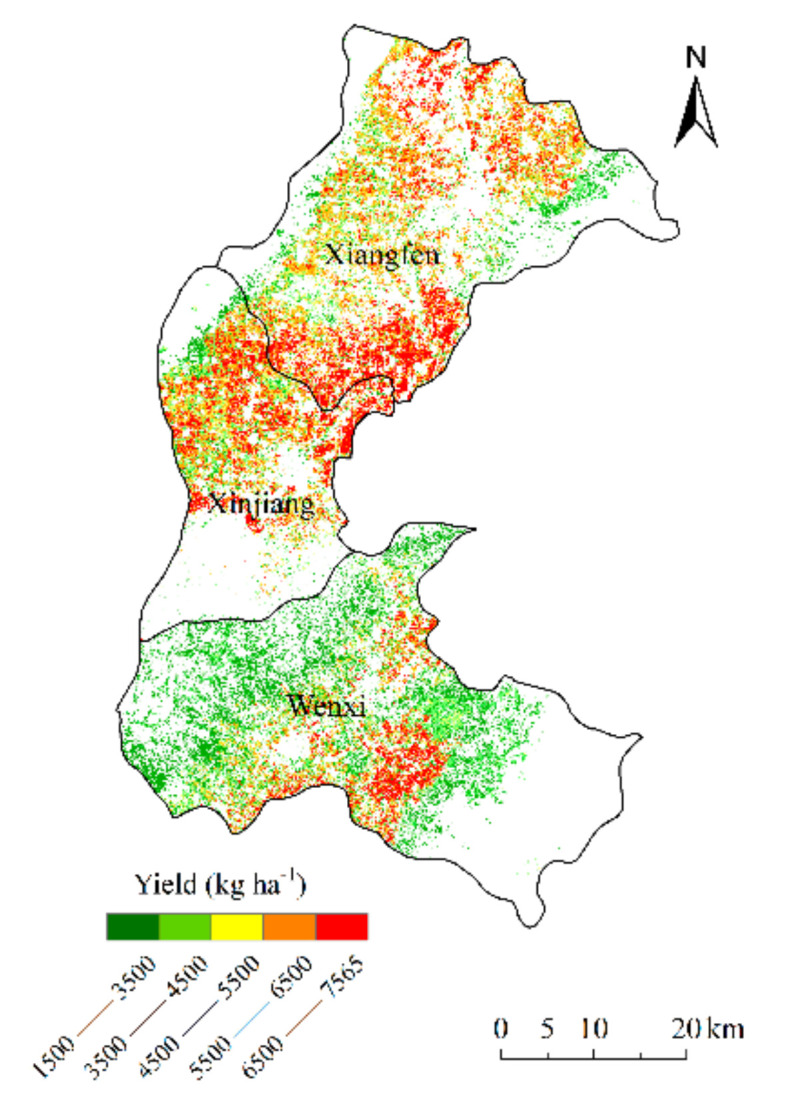
Yield distribution of winter wheat in Xiangfen, Xinjiang, and Wenxi county in 2019.

**Table 1 sensors-21-01247-t001:** Models of yield estimation for wheat using different assimilation strategies.

Wheat Planting Areas	Assimilation Strategy	Estimation Model	R^2^	*p*
Irrigated area of wheat	DA with LAI	Y = 1041.9 × LAI + 1031.4	0.61	***
	DA with soil moisture	Y = 24,505.0 × θ + 2567.6	0.59	*
	DA with soil moisture + LAI	Y = 967.2 × LAI + 4922.6 × θ + 688.6	0.61	***
Rain-fed area of wheat	DA with LAI	Y = 2105.0 × LAI − 3396.4	0.42	*
	DA with soil moisture	Y = 240,614.0 × θ − 24126.0	0.43	**
	DA with soil moisture + LAI	Y = 540.3 × LAI + 192,186.8 × θ – 20,233.8	0.49	***

Note: DA represents data assimilation; LAI represents leaf area index; * *p* < 0.05, ** *p* < 0.01, *** *p* < 0.001.

**Table 2 sensors-21-01247-t002:** Accuracy verification of the assimilation model in yield estimation of winter wheat.

Winter Wheat Planting Areas	Assimilation Strategy	RMSE (kg ha^−1^)	ARE (%)
Irrigated area of wheat	DA with LAI	427.57	6.07
	DA with soil moisture	533.64	8.49
	DA with soil moisture + LAI	436.71	6.16
Rain-fed area of wheat	DA with LAI	612.93	12.47
	DA with soil moisture	467.37	11.44
	DA with soil moisture + LAI	424.75	9.55

Note: RMSE and ARE represent the root mean square error and absolute relative error, respectively; DA represents data assimilation.

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
