# Peer review of "Estimation of Winter Wheat Yield in Arid and Semiarid Regions Based on Assimilated Multi-Source Sentinel Data and the CERES-Wheat Model"

_sensors, 2021, doi:10.3390/s21041247_

Round 1

Reviewer 1 Report

Please find my comments from the attachment, thanks!

Author Response

Response to Reviewer 1 Comments

This paper aims to estimate winter wheat yield in arid and semiarid regions using the combination of Sentinel-data and CERES-Wheat model. Authors intended to use two parameters, i.e., LAI and soil moisture, to explore their effects on the winter wheat yield estimation. However, I think this paper lacks novelty because all methods in this study are commonly used in previous study. What is the specific goal of this study?

Response: Thank you for your great comments that will effectively improve the quality of the article!

Point 1: This paper just used the widely used method to estimate winter wheat yield. What is the novelty of this study? Please clarify it.

Response 1: We consider that this paper has the following two innovations: 1) The accuracy of remote sensing parameters is an important factor that affects the accuracy of yield estimation using data assimilation-crop modeling. Currently, the commonly used data for assimilation into crop models are Landsat and MODIS data, which have a low temporal or spatial resolution. Moreover, they have limited improvement in the estimation accuracy of the data assimilation-crop modeling. There is no research on simultaneously assimilating the LAI derived from Sentinel-1 and the soil moisture derived from Sentinel-2 with high spatial and temporal resolution into crop models. 2) The existing researches have not figured out the optimal assimilation strategy for yield estimation under different water stress conditions in arid and semiarid regions. Therefore, we believe that this research is meaningful and innovative.

We have also elaborated on it in the introduction (Page 2, Lines 63-81).

Point 2: Page 1, Lines 15-32. Abstract. Why do we need to estimate crop yields in arid and semiarid regions? Please add the necessary background.

Response 2: We add the necessary background in the abstract (Page 1, Lines 15-17).

Point 3: Page 1, Lines 15-32. Abstract. All results were shown in Abstract. However, what are authors conclusions?

Response 3: We add conclusions in the abstract (Page 1, Lines 36-41).

Point 4: Page 2, Lines 54-57. I do not think these two sentences are the novelty of this paper. Suppose the Landsat and MODIS data are good enough for the crop yield estimations (e.g. high accuracy), why do we need to use Sentinel-1 and Sentinel-2?

Response 4: Apologies for the incomplete explanation of the advantages of Sentinel data. On Page 2, Lines 63-81, we gave a detailed explanation.

Point 5: Page 2, Lines 71-72. I am confused with these two sentences. Croplands in these regions are irrigated or not irrigated?\

Response 5: The sentences have been modified (Page 2, Lines 109-111).

Point 6: I cannot find any descriptions about the assimilation method in the Introduction section. Except for EnKF method, any other assimilation methods?

Response 6: We add data assimilation method in the Introduction section (Page 2, Lines 82-97).

Point 7: Page 3, Lines 98. All points, including red and yellow points, are field measured points. Please clarify it in the caption of this figure.

Response 7: The sentences have been modified (Page 3, Lines 133).

Point 8: Page 3, Lines 108-114. Were these parameters used in this study? If not, why did authors need to introduce these parameters? Just LAI and soil moisture are enough.

Response 8: The soil parameters were the input soil data of the CERES-Wheat model. It is not necessary to list it. According to your suggestion, I have deleted it from the paper.

Point 9: Page 3, Lines 116. LAI is an important parameter in this study, so I suggest authors provide more details for LAI measurements.

Response 9: We provide more details for LAI measurements (Page 4, Lines 147-156).

Point 10: Page 4, Lines 142-152. The validation of the extract winter wheat areas with the survey data is not enough as the spatial distributions of winter wheat may not correct. At least authors can evaluate the performance of the winter wheat area using the field measurements.

Response 10: Our team has been doing researches on winter wheat in this study area for more than ten years and is familiar with the planting and distribution of winter wheat in the research area. We have a farmland fertility evaluation database in Shanxi Province in 2014, including data on farmland distribution in the study area. Because the planting of winter wheat is only one part of the farmland, the area of winter wheat was extracted using the decision tree method. We also used manual discrimination and comparison with the distribution of farmland in the farmland fertility database, and finally used statistical data to analyze the accuracy of the extracted winter wheat area. Therefore, we believe that the extracted distribution of winter wheat in this study is reliable. The text on Page 5, Lines 186-195 has also been modified.

Point 11: Page 4, Lines 154-161. I do not think that such simple decision tree can identify irrigated and rain-fed areas very good. Can authors provide the accuracy of this classification results?

Response 11: We compared the irrigated winter wheat extracted from the decision tree with the extracted from the farmland fertility database of Shanxi Province in 2014, as shown in Figure 2b. The extraction accuracy was 81%. The irrigated winter wheat area in this study was more than the area extracted in the database. The extra irrigated winter wheat was mainly distributed in winter wheat planting areas at an altitude of 550-600m. From 2014 to 2019, many new farmland water conservancy projects have been built in the study area, and the actual irrigated area of winter wheat would also be higher than the recorded area in the farmland fertility database of Shanxi Province in 2014. Although we used a simple decision tree to divide the irrigated and rain-fed land, the threshold was set based on our field investigation experience over 10 years in this research area and the farmland fertility database of Shanxi Province in 2014. Therefore, we believe that the extracted irrigated and rain-fed areas for winter wheat are reliable. The text on Page 5, Lines 204-207 has also been modified.

Point 12: Pages 4-5, Lines 169-173. These are authors results rather than methods.

Response 12: We put the LAI derived from Sentinel-2 into the result analysis (Section 3.1).

Point 13: Page 5, Lines 199-202. Same as comment #12.

Response 13: We put the soil moisture derived from Sentinel-1 into the result analysis (Section 3.2).

Point 14: Page 5, Line 201. Sentinel-2 -> Sentinel-1.

Response 14: The text has been modified (Page 8, Lines312).

Point 15: Page 7, Lines 277-285. I believe the figure for the comparisons between simulated LAI and assimilated LAI needs to be added, which is better for readers' understanding.

Response 15: We add the figure for the comparisons between simulated LAI and assimilated LAI (Figure 5) (Page 9, Lines 341).

Point 16: Page 8, Lines 314-321. Same as comment #15.

Response 16: We add the figure for the comparisons between simulated soil moisture and assimilated soil moisture (Figure 7) (Page 10, Lines 380).

Point 17: Section 3.3. The comparison between the estimated winter wheat yield and county-level statistical data is better to show the accuracy of different strategies.

Response 17: In section 3.3, we also want to use county-level statistical data for comparative analysis. However, China’s 2020 statistical data has not been released so far, it is impossible to compare the estimated winter wheat yield with the county-level statistical data.

Point 18: Page 11, Lines 432-435. Even the water is not limited in the irrigated areas, why the accuracy of yield estimations are lower when adding the soil moisture for the assimilation?

Response 18: Adding the soil moisture for the assimilation could increase the modeled percolation, yield reduction in the irrigated areas. Or this could be an artifact of the bias in soil moisture data derived from Sentinel-1.

Point 19: Page 12, Lines 470-471. How did authors know Sentinel data can provide the higher accuracy of estimated data without the comparisons to MODIS and Landsat?

Response 19: The expression in the article was not rigorous. The statement “Compared with MODIS and Landsat data” has been deleted. The 2019 MODIS-LAI products have not yet been released when writing the manuscript, and another focus of this article was to explore the assimilation schemes that were suitable for different water stress areas. We will compare the accuracies of yield estimation using assimilating the crop model between the three data products in future research.

Point 20: Could authors add a comment to point out the future prospects of this paper in the Conclusion section?

Response 20: We add a comment to point out the future prospects of this paper in the Conclusion section (Page 14, Lines 518-522).

Reviewer 2 Report

Submitted article falls into the scope of Sensors Journal and I have found this paper as interesting. The study provides very interesting information how to use Sentinel 1 and 2 data and CERES-Wheat model for estimation of winter wheat yield in two regions.

Introduction contains 21 relatively new references which adequately describe the issue solved in this study. At the end of this chapter should be clearly defined the aims of this study (not “we examined…”, but “the aim of this study is – 1)… , 2)…”). The aims should be in accordance with conclusions.

Materials and Methods – are well described. Please, cite Fig.1a in text. It is not clear from Fig. 1a, where is your experimental area inside Loess Plateau.

Subchapter 2.3.1., page 4, line 147 – NDVI = Normalized Difference Vegetation Index

Line 149 - Why is the threshold value 0.25?

Line 151 – 108061 ha – please correct to 108,061 ha

Discussion – Discussion should be written as follow: this reference is in accordance with our… / is not in accordance with our… Your discussion looks like Introduction section.

Conclusion – should be in accordance with aims defined at the end of Introduction.

I recommend publishing this contribution with this major revision.

Author Response

Response to Reviewer 2 Comments

Submitted article falls into the scope of Sensors Journal and I have found this paper as interesting. The study provides very interesting information how to use Sentinel 1 and 2 data and CERES-Wheat model for estimation of winter wheat yield in two regions.

Thank you for your great comments that will effectively improve the quality of the article!

Point 1: Introduction contains 21 relatively new references which adequately describe the issue solved in this study. At the end of this chapter should be clearly defined the aims of this study (not “we examined…”, but “the aim of this study is – 1)… , 2)…”). The aims should be in accordance with conclusions.

Response 1: We defined the aims of this study at the end of the introduction (Page 3, Lines 115-119).

Point 2: Materials and Methods – are well described. Please, cite Fig.1a in text. It is not clear from Fig. 1a, where is your experimental area inside Loess Plateau.

Response 2: We cited Fig.1a in text (Page 3, Lines124).

Point 3: Subchapter 2.3.1., page 4, line 147 – NDVI = Normalized Difference Vegetation Index.

Response 3: Thanks for your suggestions. The error has been corrected (Page 5, Lines187).

Point 4: Line 149 - Why is the threshold value 0.25?

Response 4: Our research team has been conducting researches on winter wheat in this study area for more than ten years and is familiar with the planting and distribution of winter wheat in the research area. We have a farmland fertility evaluation database in Shanxi Province in 2014, including data on farmland distribution in the study area. By referring to the distribution of farmland in the farmland fertility database in 2014 and manual discrimination, the threshold was repeatedly adjusted and set to 0.25, which was most in line with the actual distribution of local winter wheat planting. It was also analyzed with the statistical planting area of winter wheat. We found the extraction accuracy was reliable. The text on Page 5, Lines 187-195 has also been modified.

Point 5: Line 151 – 108061 ha – please correct to 108,061 ha.

Response 5: Thanks for your suggestions. The error has been corrected (Page 5, Lines194).

Point 6: Discussion – Discussion should be written as follow: this reference is in accordance with our… / is not in accordance with our… Your discussion looks like Introduction section.

Response 6: Thanks for your suggestion, we revised section 4.1 in the discussion, and moved part of the content into the introduction.

Point 7: Conclusion – should be in accordance with aims defined at the end of Introduction.

Response 7: We have checked that the conclusions in accordance with aims defined at the end of Introduction.

Round 2

Reviewer 1 Report

Thanks for authors' efforts to address my comments. I think there are two comments that should be clarified in the paper.

1. Since the CERES-Wheat model can derive the yield, why not just use the derived yield from the crop model with the modified input parameters? The empirical relationship between yield and LAI/SM established by only 10 sample sites was not enough.

2. If authors intended to show the good performance of yield estimation by Sentinel data, I suggest authors can cite some papers to show the accuracy of yield estimation by Landsat or MODIS data from previous studies.

Author Response

Thanks for authors' efforts to address my comments. I think there are two comments that should be clarified in the paper.

Response: Thank you for your great comments that will effectively improve the quality of the article!

Point 1: Since the CERES-Wheat model can derive the yield, why not just use the derived yield from the crop model with the modified input parameters? The empirical relationship between yield and LAI/SM established by only 10 sample sites was not enough.

Response 1: When using the modified parameter-driven CERES-Wheat model to estimate yield, the average time needed to run a pixel is 10 s. The assimilation time for the studied region is 12.5 d, with low efficiency on the yield estimation at regional scale. In this study, the correlation between assimilated LAI/SM and yield was good, and both the estimation accuracy and efficiency were high. Therefore, the assimilated LAI/SM and yield were used in this study to construct the model for yield estimation. In the future research, we will explore the use of Google Earth Engine (GEE) platform for assimilation remote sensing into crop model to improve calculation efficiency at regional scale.

In this study, soil parameters of samples that were required to run the CERES-Wheat model included the soil physical and chemical parameters from a depth of 2 m at different layers (0 ~ 10 cm, 10 ~ 20 cm, 20 ~ 50 cm, 50-80 cm, 80 ~ 120 cm, 120 ~ 160 cm, 160 ~ 200 cm), in order to ensure the accuracy of the model. It is difficult to measure these soil parameters. We set up 20 sampling points for setting parameters for CERES-Wheat model, with 10 samples for irrigated areas and rain-fed areas each. In our study, the number of sampling points required for setting parameters for data assimilation-crop modeling framework was also referred to existing studies. For example, Xie et al. [1] set a total of 20 sampling points when using crop model assimilation system to estimate winter wheat yield in Guanzhong Plain. In our study, soil moisture, an important factor affecting the yield, has been fully considered in the yield estimation. The zoning of irrigated areas and rain-fed areas has been carried out and the yield estimation accuracy was high. In future experiments, more sampling points will be used to improve the relevance and credibility of the modeling.

References:

  1. Xie, Y.; Wang, P.; Bai, X.; Khan, J.; Zhang, S.; Li, L.; Wang, L. Assimilation of the leaf area index and vegetation temperature condition index for winter wheat yield estimation using Landsat imagery and the CERES-Wheat model. Agricultural and Forest Meteorology 2017, 246, 194-206, doi:10.1016/j.agrformet.2017.06.015.

Point 2: If authors intended to show the good performance of yield estimation by Sentinel data, I suggest authors can cite some papers to show the accuracy of yield estimation by Landsat or MODIS data from previous studies.

Response 2: I have modified the text according to your suggestion. We intended to add it to the introduction while an appropriate place cannot be targeted. Therefore, papers to show the accuracy of yield estimation by Landsat or MODIS data from previous studies have been added in Section 4.2 in Discussion (Page 17, Lines 460-467).

Reviewer 2 Report

From my point of view the authors did major revision according to reviewers comments. The paper is improved. I recommend to publish it in this present form.

Author Response

Thank you for your great comments that will effectively improve the quality of the article!